# Altered Cerebral Blood Flow and Potential Neuroprotective Effect of Human Relaxin-2 (Serelaxin) During Hypoxia or Severe Hypovolemia in a Sheep Model

**DOI:** 10.3390/ijms21051632

**Published:** 2020-02-27

**Authors:** René Schiffner, Sabine J. Bischoff, Thomas Lehmann, Andrey Irintchev, Marius Nistor, Cornelius Lemke, Martin Schmidt

**Affiliations:** 1Zentrale Notaufnahme, Otto-von-Guericke University Magdeburg, 39120 Magdeburg, Germany; 2Orthopedic Department, Jena University Hospital, Friedrich Schiller University, 07743 Jena, Germany; vorenus@web.de; 3Institute for Laboratory Animal Science and Welfare, Jena University Hospital, Friedrich Schiller University, 07743 Jena, Germany; sabine.bischoff@med.uni-jena.de; 4Institute of Medical Statistics, Computer Sciences and Documentation Science, Jena University Hospital, Friedrich Schiller University, 07743 Jena, Germany; thomas.lehmann@med.uni-jena.de; 5Department of Otorhinolaryngology, Jena University Hospital, Friedrich Schiller University, 07747 Jena, Germany; andrey.irintchev@med.uni-jena.de; 6Institute for Anatomy I, Jena University Hospital, Friedrich Schiller University, 07743 Jena, Germany; cornelius.lemke@med.uni-jena.de; 7Institute for Biochemistry II, Jena University Hospital, Friedrich Schiller University, 07743 Jena, Germany; martin.schmidt@med.uni-jena.de

**Keywords:** neuroprotection, cerebral blood flow, relaxin-2, relaxin receptors, sheep model, hypoxia, shock

## Abstract

Specific neuroprotective strategies to minimize cerebral damage caused by severe hypoxia or hypovolemia are lacking. Based on previous studies showing that relaxin-2/serelaxin increases cortical cerebral blood flow, we postulated that serelaxin might provide a neuroprotective effect. Therefore, we tested serelaxin in two emergency models: hypoxia was induced via inhalation of 5% oxygen and 95% nitrogen for 12 min; thereafter, the animals were reoxygenated. Hypovolemia was induced and maintained for 20 min by removal of 50% of the total blood volume; thereafter, the animals were retransfused. In each damage model, the serelaxin group received an intravenous injection of 30 µg/kg of serelaxin in saline, while control animals received saline only. Blood gases, shock index values, heart frequency, blood pressure, and renal blood flow showed almost no significant differences between control and treatment groups in both settings. However, serelaxin significantly blunted the increase of lactate during hypovolemia. Serelaxin treatment resulted in significantly elevated cortical cerebral blood flow (CBF) in both damage models, compared with the respective control groups. Measurements of the neuroproteins S100B and neuron-specific enolase in cerebrospinal fluid revealed a neuroprotective effect of serelaxin treatment in both hypoxic and hypovolemic animals, whereas in control animals, neuroproteins increased during the experiment. Western blotting showed the expression of relaxin receptors and indicated region-specific differences in relaxin receptor-mediated signaling in cortical and subcortical brain arterioles, respectively. Our findings support the hypothesis that serelaxin is a potential neuroprotectant during hypoxia and hypovolemia. Due to its preferential improvement of cortical CBF, serelaxin might reduce cognitive impairments associated with these emergencies.

## 1. Introduction

The human hormone relaxin-2 mediates cardiovascular adaptations during pregnancy and, particularly, systemic vasodilatation [1]. Recombinant human relaxin-2 (serelaxin^®^, Novartis Pharma, Basel, Switzerland) mediates systemic hemodynamic changes and increases renal blood flow by reducing systemic intravascular resistance concomitant with an increase in arterial compliance [2,3,4,5,6]. Moreover, serelaxin induces nitric-oxide-mediated vasodilatation [7] and can particularly reduce the infarction size in the cerebral cortex [8,9,10]. Furthermore, an intravenous bolus injection of serelaxin modulates sustained endothelial vasodilatory function and increases cerebral blood flow (CBF) in sheep [11]. All of the above-mentioned findings suggest a neuroprotective effect of relaxin-2/serelaxin. 

While the hypothesis that serelaxin might be beneficial in acute heart failure was proved wrong [12,13], serelaxin exhibits no negative impact on hemodynamic effects under normal conditions or during hypoxia or hypovolemia [14,15,16]. Currently, there is a lack of specific neuroprotective strategies for hypovolemia [17]. 

Nonetheless, studies on the neuroprotective influence of serelaxin on the CBF have not been performed up until now. CBF can be assessed using laser Doppler flowmetry, which, based on the non-invasive quantification of blood flow velocity [18], allows the differentiation of area-dependent changes in microperfusion in the cortex and subcortex. 

This study intended to investigate the influence of serelaxin on the CBF and the microcirculation under critical conditions by using two different damage systems to induce neuronal damage: either hypoxia with an O_2_ inspiration ratio of 5% for 12 min, or severe hypovolemia with a total blood loss of 50% volume. To evaluate the sustained neurological damage, we assessed the cerebrospinal fluid for the concentrations of the neuroproteins S100B and neuron-specific enolase (NSE).

## 2. Results

### 2.1. Blood Parameters

At the start of the experiment, arterial blood parameters were within their physiological ranges. Most parameters did not significantly change during the experiments (Figure 1). During hypoxia, oxygen partial pressure (pO_2_) and saturation (sO_2_) (Figure 1C,D) decreased significantly, whereas lactate, glucose, and potassium (Figure 1F,G,I) increased significantly in both treatment groups. All parameters returned to baseline within 8 min after the termination of hypoxia, except for glucose.

During the hypovolemia experiments, a significant decrease in pH (Figure 2A) and a significant increase in glucose (Figure 2G) were observed in both groups. There was a transient reduction in pO_2_ exclusively in the serelaxin group (Figure 2C), and a decrease in base excess (BE) in the control group only (Figure 2E). Lactate increased in the control group but not in the serelaxin group (Figure 2F). 

Consolidated, these data show that the experimental setting is suitable for the detection of the specific effects of serelaxin.

### 2.2. Effects of Hypoxia on Vital Parameters and CBF

In both the control and serelaxin experimental groups, O_2_ saturation, as measured by pulse oximetry, decreased rapidly to 38% at the 7 min mark of the hypoxia period (Figure 3A) (*p* < 0.001), and increased back to baseline levels after 4 min of reoxygenation. 

Hypoxia induced an average increase in the mean arterial blood pressure (MABP) in the control group (*p* < 0.02) (Figure 3B), which subsequently decreased during reoxygenation (*p* < 0.001). Over the course of the observation period, no significant alterations in the MABP were found in the serelaxin-treated group, and there were no differences between the groups. 

Hypoxia increased heart rates (HRs) above baseline in both groups: 3.5 bpm per min in the serelaxin group (*p* < 0.001) as compared to 2 bpm per min in the control group (*p* = 0.006), resulting in a mean difference of 22 bpm (*p* = 0.017) (Figure 3C). HR decreased to the baseline level during the reoxygenation and resting periods in both experimental groups (*p* < 0.01), without significant differences between the groups.

Renal blood flow (RBF) remained constant during hypoxia and reoxygenation in the control group (Figure 3D). In the serelaxin-treated group RBF showed a transient increase from baseline during hypoxia (*p* < 0.05). During reoxygenation, RBF decreased to baseline at the end of the observation period (*p* < 0.001). 

We confirmed the dichotomy of cortical and subcortical CBF in control animals during hypoxia (Figure 4A) [19]. The subcortical CBF increased over the course of hypoxia, whereas the cortical CBF decreased in comparison to baseline values (both *p* < 0.001), resulting in a significant difference in CBF between the two brain regions (*p* < 0.05). During reoxygenation, in both brain regions, CBF returned to pre-intervention values with slightly different kinetics.

Serelaxin treatment had no effect on baseline CBF but prevented the decline in cortical CBF during hypoxia (Figure 4B). Reoxygenation caused a transient increase in cortical CBF before values returned to baseline. The significant difference in the CBF in the two brain regions during hypoxia was unaltered in the presence of serelaxin (*p* < 0.001). The increases in CBF of serelaxin-treated animals in comparison to control animals were significant for both brain regions (both *p* < 0.05) (Figure 4C,D). Taken together, serelaxin improved CBF in the cortex and subcortex under hypoxic conditions.

### 2.3. Effects of Hypovolemia on CBF and Vital Parameters

In both experimental groups, O_2_ saturation, as measured by pulse oximetry, remained unchanged over the course of the experiments (Figure 5A). 

Controlled hemorrhage induced a decrease of the MABP, from 72 ± 2 mmHg at baseline to 14 ± 5 mmHg at the end of the blood removal and subsequent shock-phase in the control group (*p* < 0.001, Figure 5B). Reperfusion restored baseline MABP at the end of the observation period (*p* < 0.001). Serelaxin treatment did not alter the MAPB response during the experiments. Here, the MABP decreased from 77 ± 4 mmHg at baseline to 10 ± 2 mmHg at the end of blood removal (*p* < 0.001, Figure 5B); reperfusion similarly restored baseline MABP (*p* < 0.001).

In the control group, HR increased from 85 ± 8 bpm to 135 ± 8 bpm during blood removal (*p* < 0.001, Figure 5). During reperfusion HR decreased (*p* < 0.001), without reverting to baseline at the end of the observation period. Serelaxin treatment increased HR more moderately from 83 ± 6 bpm to 111 ± 17 bpm during blood removal (*p* < 0.001, Figure 5C), followed by a non-significant decrease at the end of the observation period. No significant differences could be observed between groups.

Hemorrhage induced a massive decrease in RBF in both groups and reperfusion increased RBF in both groups towards the end of the observation period (all, *p* < 0.001). However, there was a significant difference in the RBF response during reperfusion and recovery between the groups (*p* < 0.05, Figure 5D)—baseline RBF was only restored in serelaxin-treated animals. 

In control animals, hypovolemia (50% blood loss) had a differential effect on cortical and subcortical CBF [20]; baseline levels of cortical CBF were maintained up to a blood loss of 10%, but cortical CBF subsequently decreased to 41% ± 8% of the baseline value (*p* < 0.001, Figure 6A). In contrast, subcortical CBF remained constant up to 20% blood loss, with a subsequent decrease to 83% ± 6% of the baseline value at 50% blood loss (*p* < 0.001, Figure 6A). The effects of blood loss on cortical and subcortical CBF differed significantly in the control group during blood removal/shock (*p* < 0.001) and during reperfusion and recovery (*p* < 0.05). Subcortical CBF increased faster and transiently exceeded the baseline level with 126% ± 15% at the end of reperfusion, whereas cortical CBF returned to baseline values. 

In the serelaxin-treated group, cortical and subcortical CBF responded similarly to blood removal, shock, and reperfusion (Figure 6B). For both brain regions, the decrease in CBF during blood removal/shock was significant (*p* < 0.001). Minimum values were measured 1 min after the end of blood removal and were higher than in controls in the cortex (62% ± 4%) and lower than in controls in the subcortex (73% ± 7%). Cortical and subcortical CBF increased during reperfusion and throughout recovery, both exceeding baseline values (maximum values were 123% ± 6% in the cortex and 114% ± 5% in the subcortex), but statistical significance was missing because the linear mixed models were calculated using baseline values as the reference group. 

A direct comparison between CBF changes in the control group and the serelaxin-treated group during hypovolemia revealed a significant improvement in cortical CBF in the presence of serelaxin (*p* < 0.001, Figure 6C). The mean improvement in CBF during hypovolemia amounted to 21%. In contrast, subcortical CBF decreased in the serelaxin-treated animals in comparison with control animals during blood removal/shock (*p* = 0.002) and during reperfusion/recovery (*p* < 0.05) (Figure 6D).

### 2.4. Effects of Hypoxia and Hypovolemia on Neuroproteins

In the hypoxia control group, the cerebrospinal fluid concentration of S100B increased from 60 ng/mL (all data are medians) to 127 ng/mL (*p* = 0.031), and NSE remained nearly constant (4 ng/mL versus 3 ng/mL) during hypoxia (Figure 7A,B). In the serelaxin-treated hypoxia group, the cerebrospinal fluid concentration of S100B decreased from 101 to 26 ng/mL (*p* = 0.031), while the NSE concentration was reduced from 5 to 1 ng/mL (*p* = 0.063). Obviously, there was a (tendency for a) reduction in both neuroproteins in the serelaxin-treated group in comparison with the control animals after hypoxia, albeit statistical significance was missing for NSE.

In the hypovolemic control group, the cerebrospinal fluid concentration of S100B increased from 107 ng/mL (all data are medians) to 225 ng/mL (*p* = 0.063), and NSE increased from 13 to 35 ng/mL (*p* = 0.031) (Figure 7C,D). In the serelaxin-treated hypovolemic group, the cerebrospinal fluid concentration of S100B decreased from 155 to 95 ng/mL (*p* = 0.031), while the NSE concentration decreased from 15 to 3 ng/mL (*p* = 0.063). After hypovolemia, we observed significant reductions in both neuroprotein levels in the serelaxin-treated group as compared with the control group (S100B, *p* = 0.041; NSE, *p* = 0.002; Mann–Whitney rank sum-test).

In addition, cortical sections from serelaxin hypoxic, control hypoxic, and control non-hypoxic animals were stained for NeuN and S100B and counterstained with a Nissl-like stain. As revealed by cell morphology, immunostaining was restricted to neurons (NeuN) and astrocytes (S100B) (Figure 8A–F). There were no apparent pathological alterations, and staining intensities were similar in all three of the analyzed groups (Figure 8G). These findings suggest that the 12 min episode of hypoxia that was utilized (i.e., the observation period) was too short to induce immediately detectable morphological tissue pathology.

### 2.5. Expression of Relaxin Receptors and Potential Downstream Signaling Proteins

Brain vessel samples from the cortex and subcortex were harvested after euthanasia from 12 age-matched control sheep that did not undergo the experimental procedure. Western blotting revealed the expression of receptors for relaxin-2, RXFP1 and RXFP2 (Figure 9A), at the protein level in both brain regions under investigation. A molecular weight of approximately 85 kDa for the higher molecular weight band detected by the anti-human RXFP1-antibody corresponds to the full-length receptor. An additional band of approximately 65 kDa was detected with the anti-RXFP1 antibody, which was not seen in a previous study using brain samples not enriched for blood vessels [11]. For RXFP2, only bands with lower molecular masses (70 and 55 kDa) than the expected 85 kDa were detectable, which is consistent with the findings of the aforementioned study [11]. For all RXFP variants, significantly higher amounts were detected in brain vessel samples from the subcortex as compared with the cortex (Figure 9B). 

The immunohistochemistry analysis, which utilized the same antibodies for RXFP1 and RXFP2 (with and without antigen retrieval), detected no immunolabeling above background levels in the paraffin sections, formaldehyde-fixed sections, or fresh-frozen sections of the frontal cortex or thalamus.

We hypothesized that serelaxin action on brain vessels should lead to alterations in the activities of key signaling mediators of relaxin receptors. Indeed, serelaxin treatment reduced the Ser1177 phosphorylation of eNOS (Figure 10A) and increased the ratio of a truncated variant of nNOS (nNOS^149^) over the normal/long variants (nNOS^161/165^) (Figure 10B). In addition, the phosphorylation of CREB (cAMP-responsive element-binding protein) variants tended to be reduced after serelaxin treatment (Figure 10C). These findings confirm that serelaxin exerts an effect on brain vessels. Furthermore, serelaxin treatment led to differences in activation-state related phosphorylation of predominantly ERK2, suggesting differential relaxin signaling in cortical versus subcortical vessels (Figure 10D).

## 3. Discussion

The key finding of this study is that serelaxin can alter region-specific CBF. Furthermore, our results imply a neuroprotective effect during hypoxia and hypovolemia in a sheep model.

We observed similar blood gas levels during hypovolemia in both groups. The pulsatile oxygen saturation and the blood gases exhibited a similar response pattern during the observation period in both the hypoxic and the hypovolemic groups. Consequently, the comparability of the experimental procedure of the respective experimental groups can be affirmed.

Both the MABP and the HR increased significantly during hypoxia in both groups, as already evidenced by other studies in human subjects [21]. In our experimental study, the impact on HR was more pronounced. The HR increase was more significant in the serelaxin-treated group, which correlates with both the known properties of the substance and the results of another study in a sheep model [11]. The positive chronotropic effects of relaxin-2 have been described previously [14]. While RBF increased only moderately during hypoxia in the control group over the course of the observation period, a significant increase was observed in the serelaxin-treated group. Moreover, this finding suggests that serelaxin affects the renal circulation in line with the known function of relaxin as a very potent and rapid (within minute) arterial vasodilator [11,22]. In regard to the RBF, the large variance within groups can be explained by the large inter-individual differences in renal blood flow in sheep, which, in our measurements, varied between 40 and 100 mL/min in the arteria renalis, even before the experimental intervention. 

The increase in HR, however, can most likely be ascribed to the high dosage of serelaxin utilized in the experiments. In contrast to clinical application schemes for the treatment of heart failure [23], we utilized a bolus injection to facilitate the detection of weak responses on the CBF. In addition, a bolus injection would be preferable in an emergency setting, such as hypoperfusion, in which the objective would be quick restoration of the blood flow in the brain. The equivalent of a daily human dose [23] was administered to the animals, which almost immediately ensured a serum level triggering response. With regard to the solvent of serelaxin (20 mM sodium acetate, pH 5.2) an effect can be excluded, as the applied quantity translates into an increase of less than 0.1 percent of the physiological blood concentration in sheep [11,24].

Sheep were elected as the experimental model for our studies because of their similarities to humans, including body weight, physiological parameters, and organization of the brain. Using this model, we have previously shown that hemorrhage [20] or hypoxia [19], respectively, trigger differential CBF responses in the cortex and subcortex. Both damage models resulted in lower cortical CBF in comparison with subcortical CBF, which was replicated in this present study. In addition, studies in small animal species have previously led to conflicting results, although some more recent studies in conscious rats have also shown that CBF decreased during hypoxia in most of the cortical regions and that CBF in the subcortical and brain stem structures remained unchanged during hemorrhage [25,26]. One study described different autoregulatory responses in the cerebral cortex and subcortex and highlighted a higher subcortical CBF [27]. However, using various experimental paradigms, no decisive conclusion as to differential CBF under clinically relevant conditions could be drawn from the results of small animal studies (reviewed in [26] and discussed in more detail in [11]). 

Serelaxin treatment significantly increased CBF in the cortex and subcortex during hypoxia. It completely prevented the cortical CBF reduction found in control animals, and it triggered a significantly stronger increase in subcortical CBF. Similarly, serelaxin significantly improved cortical CBF during pronounced hypovolemia, whereas it concomitantly decreased subcortical CBF. This improvement in cortical CBF was mirrored by a higher demand for oxygen (reduced pO_2_) and the lack of lactate increase in the serelaxin-treated animals, indicating better preservation of aerobic energy generation during hypovolemia. As CBF was the only parameter of blood flow that was altered in response to serelaxin treatment, this suggests that it is indeed the cerebral cortex whose energy supplies are improved by serelaxin. In addition, in a previous study, we found that serelaxin triggered an increase in CBF in the cortex but not in the subcortex of sheep under normal conditions [11]).

The varying degrees of CBF responses during controlled hypoxia and hemorrhage reveal differences in the autoregulation of the cerebral cortex and subcortex. This differs from the previous assumption that the centralization of the blood circulation protects the entire brain during adult life [21,28,29]. According to our investigations, serelaxin alters the area-dependent distribution of the CBF in favor of the cerebral cortex. Furthermore, examination of the cerebrospinal fluid revealed significantly lower levels of S100B and NSE, in contrast to the tendency for increased neuroprotein concentrations in both groups, which can be attributed to the experimental setup, for example, the introduction of the laser Doppler probes. The biomarkers S100B and NSE have major clinical relevance for the estimation of neurological damage [30,31,32]. In particular, the timescale for the increase in neuroprotein concentrations caused by neuronal damage is much shorter than damage detection by clinical routine imaging technologies. Furthermore, they can be used as predictors of cognitive dysfunction in various clinical contexts [33,34]. In summary, the observed decline in the neuroprotein levels implies a neuroprotective effect of serelaxin [35]. However, this effect was not detectable immunohistochemically, which is likely due to the high sensitivity of cerebrospinal fluid neuroproteins for neuronal damage. 

The insufficient protection of the cortical CBF during hypoxia and hypovolemia may contribute to the higher vulnerability of the cerebral cortex to ischemic brain damage during hemorrhage [36,37]. Our results highlight that maintenance of the CBF is essential in managing hypoxic or bleeding patients in order to improve neurological outcomes [36,37,38]. Furthermore, the relaxin-2-preparation serelaxin may provide a means for diminishing neuronal damage. It might be suitable for application in emergency scenarios or in intensive care settings. However, the results of the RELAX-AHF-studies, which initially found long-term benefits of serelaxin in the treatment of acute heart failure [39], which ultimately could not be confirmed [13], indicate that independent confirmation of our findings is necessary.

The different effects of serelaxin treatment on the subcortical and cortical CBF suggest that region-specific control of the CBF is a consequence of unequal relaxin receptor densities in the corresponding areas. RXFP1 and RXFP2 represent the known relaxin-2-activated receptors [40]. These receptors are involved in various aspects of the remodeling of the cerebral parenchymal arterioles [41,42] as well as in the reduction of vascular resistance and in an increased blood flow in renal and systemic small arteries in humans and rats [1,43]. These effects, however, often require longer timescales. We used Western blotting to detect these receptors in brain vessel samples from the frontal cortex and the subcortex (thalamus). Relaxin-2 has roughly nanomolar affinities for both receptors [44], the affinity being about an order of magnitude higher for the RXFP1 receptor. Considering that only the full-length RXFP1 protein was detectable by Western blotting, it can be assumed that the RXFP1 receptor is responsible for transducing the serelaxin signal. Interestingly, the average expression levels of all detected RXFP protein variants were higher in subcortical vessels than in cortical vessels. These observations are nonetheless only based on Western blotting experiments. Unfortunately, the immunohistochemical analysis of RXFP1 and RXFP2 in the sheep tissues was hampered by the unsuitability of the antibodies. 

Despite the lower levels of RXFP expression in the cortical vessels, fast serelaxin-induced changes in the cerebral microcirculation were observed. Therefore, the quick response of the cortical cerebral microcirculation in response to serelaxin treatment most likely results from area-specific differences in signal transduction pathways activated through RXFP1 receptors by the agonist, rather than from receptor expression levels. The immediate response of relaxin-2 signaling in blood vessels involves the activation of phosphoinositide 3-kinase via G_βγ_, activation of protein kinase B/Akt, phosphorylation of endothelial NO synthase (eNOS), and release of NO (extensively reviewed in [40,44]). Additional hallmarks of RXFP1 signaling are the activation of protein–kinase A and MAP–kinase pathways. Our analysis of the expression and activation state of key signaling proteins reveals serelaxin-induced alterations (however, samples were prepared from animals not undergoing the experimental procedure). Therefore, region-specific differential activation of one of these mediators is most likely responsible for the observed differences in the CBF. From the finding of higher levels of RXFP expression in the subcortical vessels, a working hypothesis for future studies can be deduced: relaxin-2 or a related relaxin-family type of ligand in combination with RXFP1 and/or RXFP2 may mediate the superior autoregulation of the subcortex as compared to the cortex.

## 4. Materials and Methods 

### 4.1. Animal Care and Surgical Instrumentation

All procedures were approved by the Thuringia Animal Welfare Committee (located in Bad Langensalza; permission number: TVV 02-056/13; date of approval 13 December 2013) and conducted in accordance with the ARRIVE guidelines. Twenty-four 2–6 year old female Merino long wool sheep, weighing 87.3 ± 10.1 kg, underwent surgery in accordance with the Guide for the Care and Use of Laboratory Animals [45]. After food withdrawal for 24 h, anesthesia was induced by intramuscular injection of 10–15 mg/kg ketamine (Ketamin-Hydrochlorid^®^, Pfizer, Berlin, Germany) and 0.2 mg/kg midazolam (Midazolam-Hameln^®^, Hameln Pharmaceuticals, Hameln, Germany). For analgesia, 0.002 mg/kg fentanyl (Fentanyl, Janssen, Neuss, Germany), and for muscular relaxation 0.05 mg/kg pancuronium (Pancuronium-Actavis^®^, Actavis, München, Germany), was administered intravenously. After orotracheal intubation, anesthesia was maintained via inhalation of 1.5% isoflurane (Isofluran–Actavis^®^, Actavis, Langenfeld, Germany) in 50% oxygen over the course of the entire experiment. All sheep were instrumented with vascular catheters (Arterial Leadercath, Vygon, Aachen, Germany), which were inserted into the carotid artery for blood sampling and blood pressure measurement; into the left jugular vein (Trilyse Expert^®^, Vygon) for intraoperative administration of analgesics; and into the right jugular vein (Corodyn P2 F6, Braun Melsungen AG, Melsungen, Germany) for sampling from the arteria pulmonalis. After a left lateral laparotomy, we inserted a flow probe (Animal Blood Flowmeter T 206, Transonic, Ithaca, NY, USA) around the left kidney artery to determine the renal blood flow (RBF). Skin and epicranial aponeuroses were removed from the skull, and two borehole trepanations of 1 cm diameter were performed at 2 and 4 cm in front of the interauricular line and 1 cm lateral of the midline. 

### 4.2. Cerebral Blood Flow and Microcirculation

Single fiber laser Doppler flow probes (diameter 400 µm, Moor, Devon, UK) were inserted 2 mm into the parietal cortex and 2.7 cm into the subcortex (thalamus) for the continuous monitoring of capillary CBF changes [18]. The CBF was recorded at a sampling rate of 40 sec^−1^ using a laser Doppler flowmeter (DRT4, Moor, Devon, UK). 

### 4.3. Experimental Protocol

After 20 min of baseline recordings (baseline 0), a bolus injection of 20 mL saline (Isotonische Kochsalzlösung^®^, Fresenius Kabi, Bad Homburg, Germany) was administered to twelve sheep (6 sheep in hypoxic and 6 sheep in hypovolemic control groups, respectively); while 30 µg/kg of serelaxin (0.1–0.15 mL stock solution) diluted in saline (Serelaxin®, Novartis Pharma, Basel, Switzerland) was administered to the other twelve sheep (6 sheep in hypoxic and 6 sheep in hypovolemic serelaxin groups). After another 10 min of baseline recordings, hypoxia or hypovolemia was induced, as described below. Cortical and subcortical CBF, mean arterial blood pressure (MABP), electrocardiogram (ECG), RBF, and oxygen saturation were continuously recorded. MABP was recorded with a transducer (Combitrans Transducer, Braun, Melsungen, Germany). ECG was derived using intracutaneous wire electrodes. All biophysical variables were amplified and sampled continuously at 1000 Hz using a data acquisition and analysis system (Labchart Pro7, ADInstruments, Spechbach, Germany). Heart rate (HR) was triggered from R waves using Labchart Pro 7.

### 4.4. Induction and Resolution of Hypoxia

After baseline recordings, hypoxia was induced by inhalation of 5% oxygen and 95% nitrogen and was maintained for 12 min. After this period, reoxygenation was started with 50% oxygen over 20 min under constant monitoring. 

### 4.5. Induction and Resolution of Hypovolemia

After baseline recordings, a controlled severe hemorrhage was induced in 12 sheep through the removal of 50% of the estimated total blood volume, which approximates 7% of total body weight in sheep [46]. The blood was removed with a continuous flow rate in order to remove the estimated blood volume within 20 min. To simulate a shock phase, the animals were kept at 50% blood volume for 20 min, followed by reperfusion of the removed blood over 20 min, and subsequent measurements for another 15 min. The blood was conserved in empty infusion bags (Isotonische Kochsalzlösung, freeflex^®^, Fresenius Kabi, Bad Homburg, Germany) with the addition of 1000 IE heparin (Heparin-Natrium 5000, Ratiopharm, Ulm, Germany). 

### 4.6. Analysis of Blood Gases

Blood samples for analysis of blood gases, ions (sodium, potassium, calcium and chloride), glucose, and lactate were taken at various timepoints (hypoxia: before oxygen restriction, every other minute during oxygen restriction, and 8 min after start of reoxygenation; hypovolemia: before and at the end of blood removal, after reperfusion, and at the end of measurements). The samples were measured on a standard clinical blood gas analyzer (ABL 600, Radiometer GmbH, Willich, Germany).

### 4.7. Quantitation of S100B and NSE

Cerebrospinal fluid was collected from the cisterna magna at the end of baseline measurements and at the end of recordings, thereby covering the complete hypoxic and hypovolemic episodes, respectively. S100B and NSE (neuron-specific enolase) levels were measured using routine clinical laboratory methods.

### 4.8. Sample Preparation and Western Blotting

The animals were euthanized by intravenous injection of pentobarbital sodium (Narcoren, Merial, Halbergmoos, Germany) after the recordings. Representative brain vessel samples were snap frozen in liquid nitrogen. Sample preparation and Western blotting were done as described previously [11], with the exception that instead of total brain tissue, which was used in the previous study, brain vessels were analyzed in this study (full size blots are available in the Appendix A). In brief, samples were homogenized in 1 mL RIPA-buffer using a S2 advanced focused acoustics system (Covaris, Woburn, Massachusetts, USA) in frequency sweeping mode (duty cycle, 20%; intensity, 8; cycles/burst, 200; duration, 300 s). Fifty micrograms of protein per sample was separated in 10% SDS-gels [47] and semi-dry blotted onto Immobilon-P membranes [48]. Antibodies used for the detection of proteins were rabbit anti-relaxin receptor 1 (RXFP1 (H-160), 1:250; sc-50328), rabbit anti-relaxin receptor 2 (RXFP2 (H-150), 1:250; sc-50327), rabbit anti-NO synthase 1 (nNOS (H-299), 1:1000; sc-8309), rabbit anti-NO synthase 3 (eNOS (H-159), 1:1000; sc-8311), goat anti-rabbit-IgG-HRP (1:10.000; sc-2004), goat anti-mouse-IgG-HRP (1:5.000; sc-2031) (all from Santa Cruz Biotechnology; Dallas, TX, USA), rabbit mAb anti-phospho-Ser1177-eNOS ((C9C3), 1:1000; CST-9570), rabbit mAb anti-p44/42 MAPK (ERK1/2) ((137F5), 1:1000; CST-4695), rabbit mAb anti-phospho-Thr202/Tyr204-p44/42 MAPK (P-ERK1/2) ((D13.14.4E), 1:1000; CST-4370), rabbit mAb anti-CREB ((48H2), 1:1000; CST-9197), rabbit mAb anti-phospho-Ser133-CREB ((87G3), 1:1000; CST-9198) (all from Cell Signaling Technology; Frankfurt am Main, Germany), and mouse anti-β-actin (1:5.000; AC-15, A5441; Sigma-Aldrich, Taufkirchen, Germany). Signals were generated with the Western Lightning Ultra chemiluminescence-kit (PerkinElmer; Rodgau, Germany) for all proteins except β-actin, where a homemade enhanced chemiluminescence solution was used. Detection was done in a GBox iChemiXL (Syngene, Cambridge, UK). A reference sample was included in each gel (produced by mixing nine brain vessel extracts) for the comparison of protein expression in different gels. Quantitation was done with the normalized (against the reference sample) band intensities and presented as the ratio of the signal of interest over the β-actin signal, which served as the housekeeping control. Similarly, phospho-protein signals were normalized to total protein signals.

### 4.9. Immunofluorescence

Immunohistochemistry was performed using frontal cortex samples fixed in 10% formalin for 24 h at room temperature (RT) and embedded in paraffin. Sections (6 µm thickness) were obtained on a microtome and collected on glass slides. Immunohistochemistry was performed according to an established protocol [49]. Tissue sections were subjected to antigen retrieval (30 min at 80 °C in 10 mM sodium citrate solution, pH 9.0). Non-specific binding was blocked for 1 h at RT with phosphate-buffered saline (PBS, pH 7.3) containing 0.2% (*v*/*v*) Triton X-100 (Sigma-Aldrich, Taufkirchen, Germany), 0.02 % (*w*/*v*) sodium azide (Sigma-Aldrich), and 5% (*v*/*v*) normal goat serum (Jackson ImmunoResearch Europe, Suffolk, UK). The primary antibodies were diluted in PBS containing 0.5% (*v*/*v*) lambda-carrageenan (Sigma-Aldrich) and 0.2% (*w*/*v*) sodium azide and were applied to the sections overnight at 4 °C. The primary antibodies used were anti-S100B (rabbit polyclonal, 1:2000, Agilent Pathology Solutions, Santa Clara, CA, USA), anti-NeuN (mouse monoclonal, clone A60, 1:1000, Chemicon, Hofheim, Germany), and anti-NSE (rabbit polyclonal, 1:100–1:10000, Polysciences Europe, Hirschberg an der Bergstrasse, Germany). After incubation with the primary antibody and PBS washes, sections were incubated in Cy3-conjugated goat anti-rabbit antibody (Jackson ImmunoResearch) diluted 1:200 in PBS containing 0.5% lambda-carrageenan and 0.2% (*w*/*v*) sodium azide, for 2 h at RT. Counterstaining was performed using green NeuroTrace fluorescence (Nissl-like) stain (Life Technologies, Darmstadt, Germany) diluted 1:200 in PBS for 20 min at RT. Staining controls included omitting the first antibody or replacing it by normal rabbit or mouse IgG (Jackson ImmunoResearch). Section analysis and documentation were performed using Stereo Investigator 8.1 software (MicroBrightField Europe, Magdeburg, Germany) and a fluorescence microscope (Axioskop 2 mot plus, Zeiss, Oberkochen, Germany) equipped with a motorized stage (Zeiss) and a CX 9000 digital camera (MicroBrightField). Intensities of immunolabeling (gray values) were measured using images (objective 40×) from cortical layers II and III and Image J 1.49 software (https://imagej.nih.gov/ij/). To ensure reproducibility, the same acquisition parameters were used for all images. Analyses were performed using sections stained for NeuN and S100B. No specific NSE labeling could be achieved at any antibody dilution tested (1:100–1:10.000).

### 4.10. Randomisation and Blinding

Allocation of animals to treatment groups was performed randomly. Group allocation was known to the in vivo experimenters because the increase of HR during the experiments necessarily unblinded group allocation. For blood gases and in vitro experiments, the evaluators were blinded to the group allocations.

### 4.11. Statistical Analysis

Linear mixed models were fitted to analyze the difference between groups over time. Furthermore, the data of an individual over time are typically not independent, which was considered in the analyses. Fixed effects were group, procedure, time, and the interaction of group and time. By analyzing the interaction term, we were able to test for measurement differences between the groups over time. We performed four linear mixed models for each outcome parameter: one with the hypovolemia data and one during reperfusion, one with the hypoxia data and one during reoxygenation. Blood parameters of the hypovolemia animals and results of in vitro tests were analyzed with Wilcoxon signed rank tests (within one treatment group) or with the Mann–Whitney rank sum test (for comparison between treatment groups and for blood parameters during hypovolemia). The significance level was set to α = 0.05. All analyses were performed with IBM SPSS Statistics 23.0. Continuous outcomes are described by mean ± SEM.

## 5. Conclusions

To conclude, the injection of a high dose of serelaxin exerts immediate effects on the cerebral microcirculation. Therefore, serelaxin might be suitable for improving cortical CBF in clinically relevant scenarios, like penumbra, intracranial hemorrhage, shock, asphyxia, and cerebral vasospasm. Interestingly, the effects on the cerebral cortex can be observed quickly after administration of the compound. These findings need to be confirmed in relevant animal models and could possibly be translated into clinical practice.

Further experiments, in particular, long-term follow-up studies, are required to validate this notion at the molecular and structural levels.

## Figures and Tables

**Figure 1 ijms-21-01632-f001:**
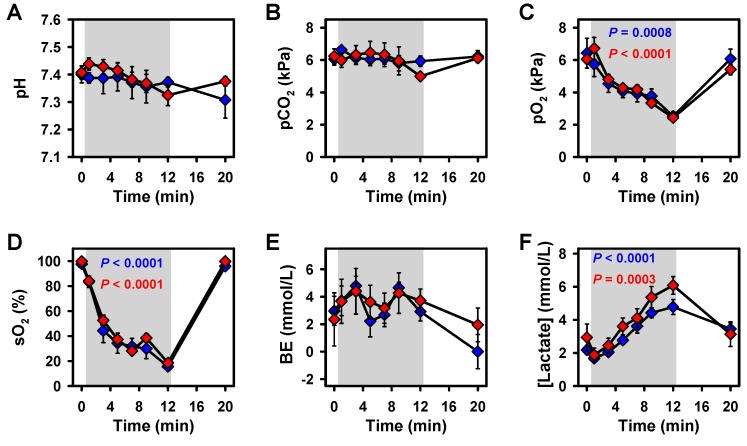
Effects of hypoxia and reoxygenation on blood parameters. Values are given for baseline (*t* = 0 min), during hypoxia (up to *t* = 12 min), and after reoxygenation (*t* = 20 min) in controls and the serelaxin-treated group. (A) pH, (B) partial pressure of carbon dioxide (pCO_2_), (C) pO_2_, (D) sO_2_, (E) base excess (BE), (F) lactate, (**G**) glucose and (**H**) sodium, (**I**) potassium, (**J**) calcium, and (**K**) chloride. Mean ± SEM; *p*-values for changes during hypoxia result from linear mixed models.

**Figure 2 ijms-21-01632-f002:**
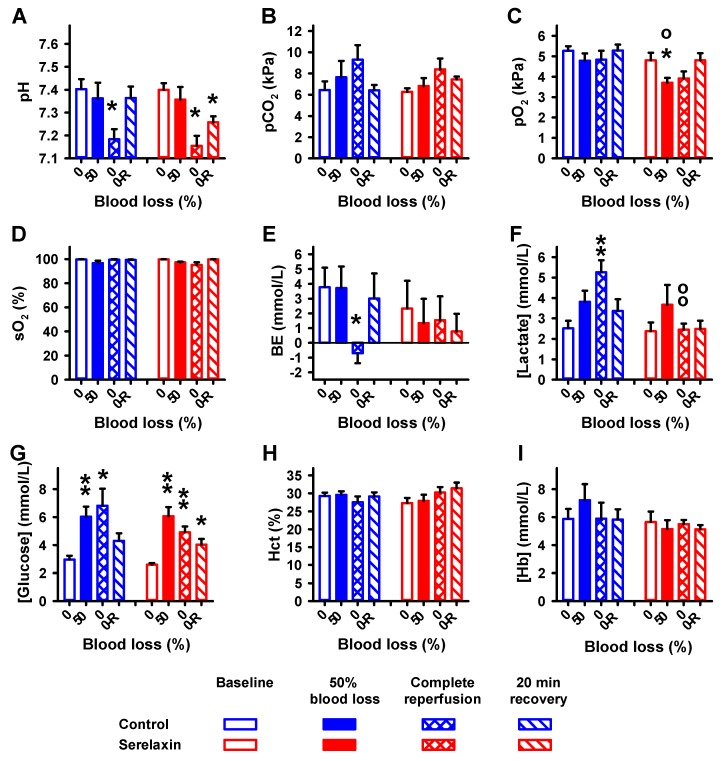
Effects of blood loss and reperfusion on blood parameters. Values are given for baseline, removal of 50% blood, complete reperfusion, and 20 min after the complete reperfusion in controls and the serelaxin-treated group. (A) pH, (B) pCO_2_, (C) pO_2_, (D) sO_2_, (E) BE, (F) lactate, (**G**) glucose, (**H**) hematocrit (Hct), and (**I**) hemoglobin (Hb). Mean ± SEM; * *p* < 0.05, ** *p* < 0.005 compared to baseline; o, *p* < 0.05; oo, *p* < 0.005 compared to controls.

**Figure 3 ijms-21-01632-f003:**
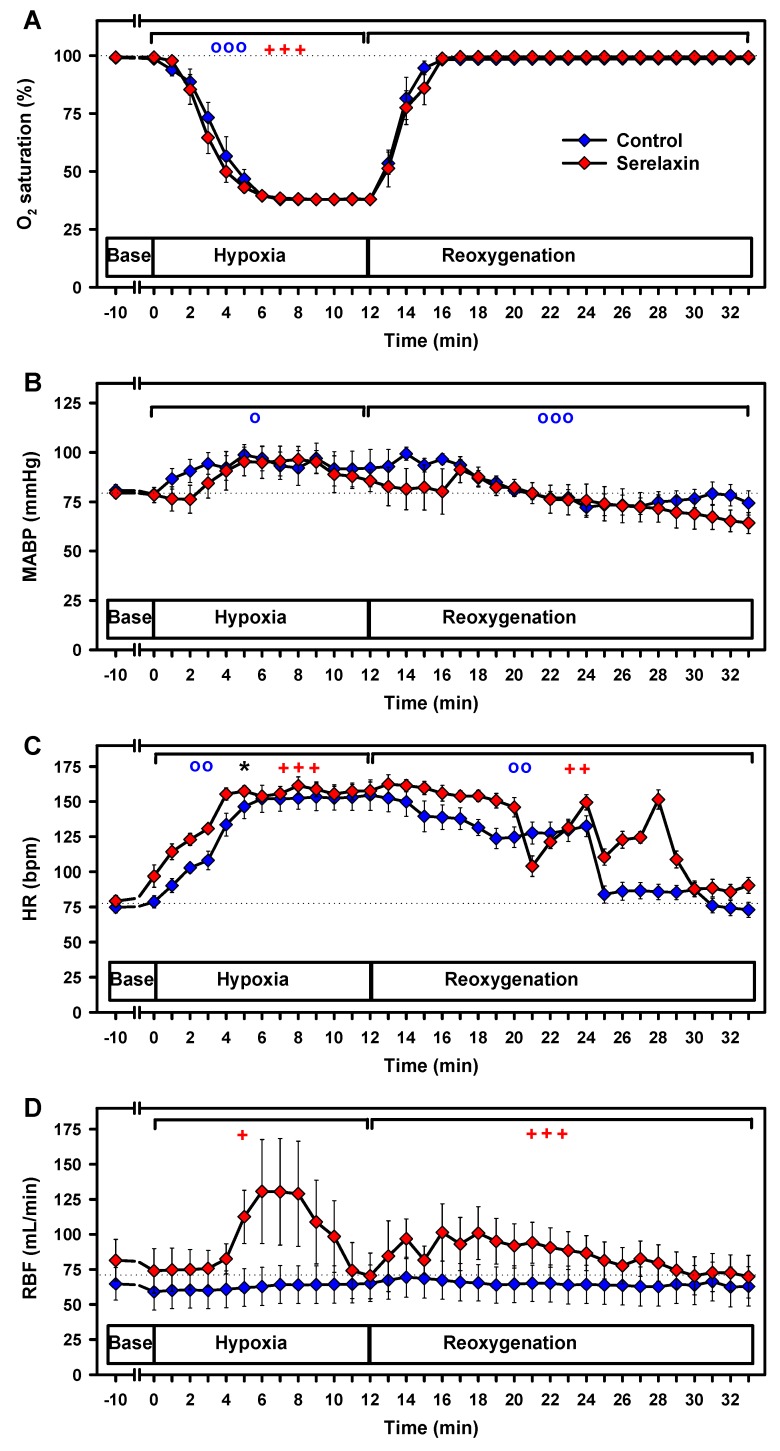
Effects of hypoxia and reoxygenation on vital parameters. (A) oxygen saturation (O_2_ saturation), (**B**) mean arterial blood pressure (MABP), (C) heart rate (HR), and (D) renal blood flow (RBF). Linear mixed models were separately calculated for hypoxia and reoxygenation versus baseline values. o, *p* < 0.05; oo, *p* < 0.01; ooo, *p* < 0.001 for the comparison of the control group with baseline values. + *p* < 0.05; ++ *p* < 0.01; +++ *p* < 0.001 for the comparison of the serelaxin-treated group with baseline values. * *p* < 0.05 for the comparison between the two experimental groups. Dotted lines indicate baseline values.

**Figure 4 ijms-21-01632-f004:**
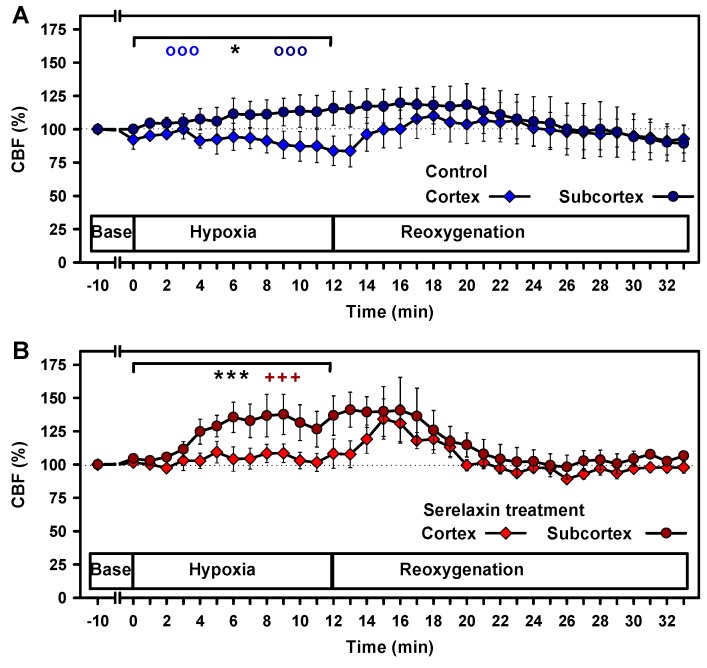
Effects of hypoxia and reoxygenation on cortical and subcortical cerebral blood flow (CBF). Comparison of cortical and subcortical CBF in the control group (A) and under serelaxin treatment (B). Comparison of controls and serelaxin treatment for the (C) cortex and (D) subcortex. Linear mixed models were separately calculated for hypoxia and reoxygenation. Statistically significant effects were found during hypoxia: ooo, *p* < 0.001 for the comparison of the control group with baseline values. +++ *p* < 0.001 for the comparison of the serelaxin-treated group with baseline values. * *p* < 0.05; *** *p* < 0.001, for the comparison of two experimental groups during hypoxia. No significant effects were detected during the course of reoxygenation. Dotted lines indicate baseline values.

**Figure 5 ijms-21-01632-f005:**
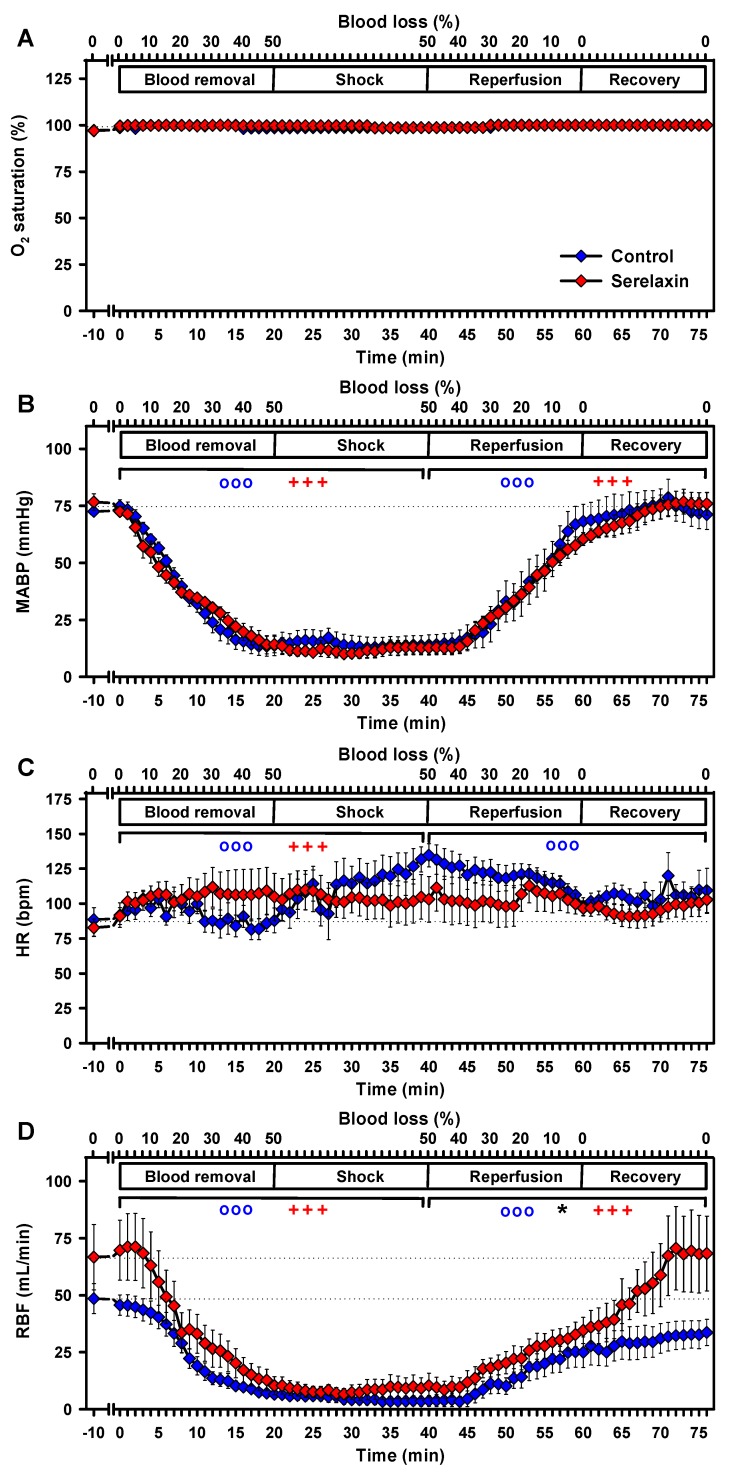
Effects of controlled hemorrhage with a resulting shock period and reperfusion on vital parameters. (A) oxygen saturation (O_2_ saturation), (**B**) MABP, (C) HR and (D) RBF. Linear mixed models were separately calculated for hypoxia and reoxygenation versus baseline values. ooo, *p* < 0.001 for the comparison of the control group with baseline values. +++ *p* < 0.001 for the comparison of the serelaxin-treated group with baseline values. * *p* < 0.05 for the comparison between the two experimental groups. Dotted lines indicate baseline values.

**Figure 6 ijms-21-01632-f006:**
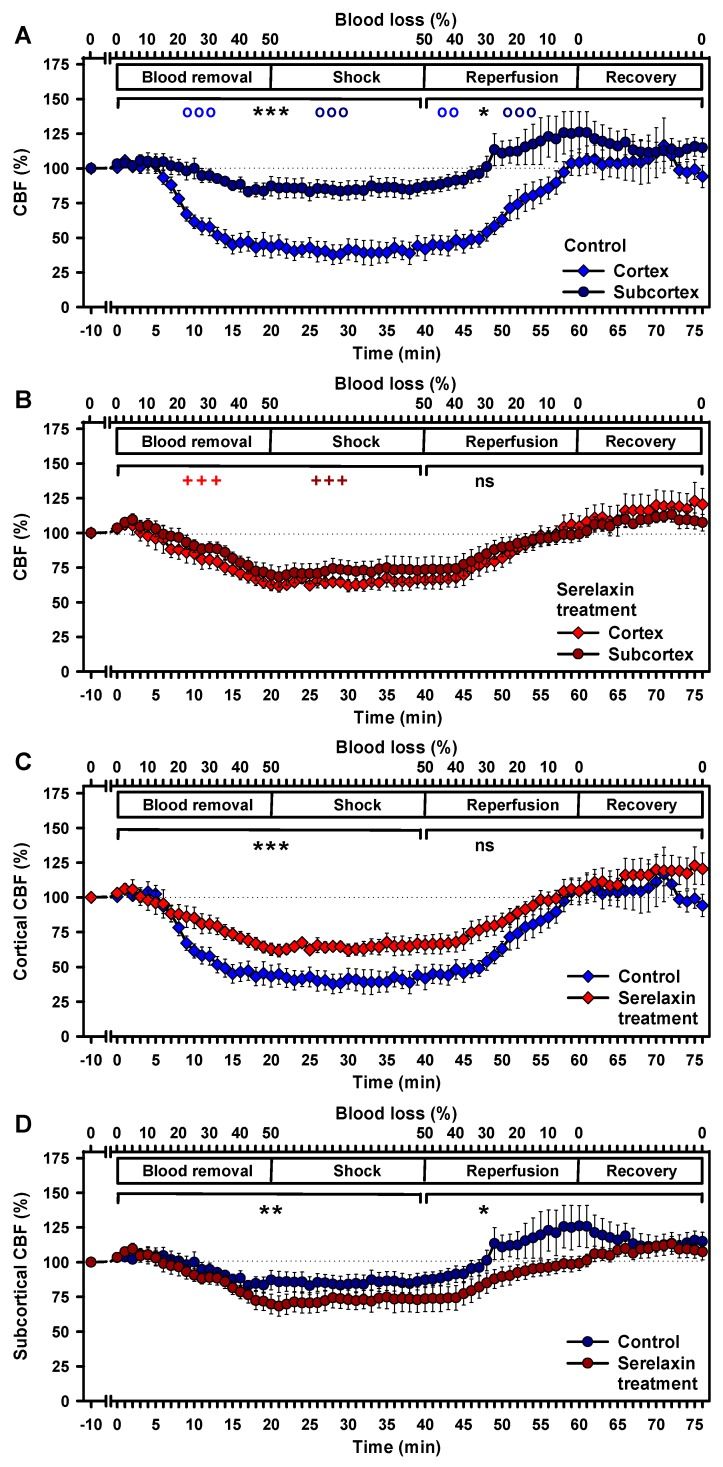
Effects of controlled hemorrhage with a resulting shock period and reperfusion on cortical and subcortical CBF. Comparison of cortical and subcortical CBF in the control group (A) and under serelaxin treatment (B). Comparison of controls and serelaxin treatment for the cortex (C) and subcortex (D). Linear mixed models were separately calculated for hypoxia and reoxygenation versus baseline values. oo, *p* < 0.01; ooo, *p* < 0.001 for the comparison of the control group with baseline values. +++ *p* < 0.001 for the comparison of the serelaxin-treated group with baseline values. * *p* < 0.05; ** *p* = 0.002; *** *p* < 0.001 for the comparison between the two experimental groups. Dotted lines indicate baseline values.

**Figure 7 ijms-21-01632-f007:**
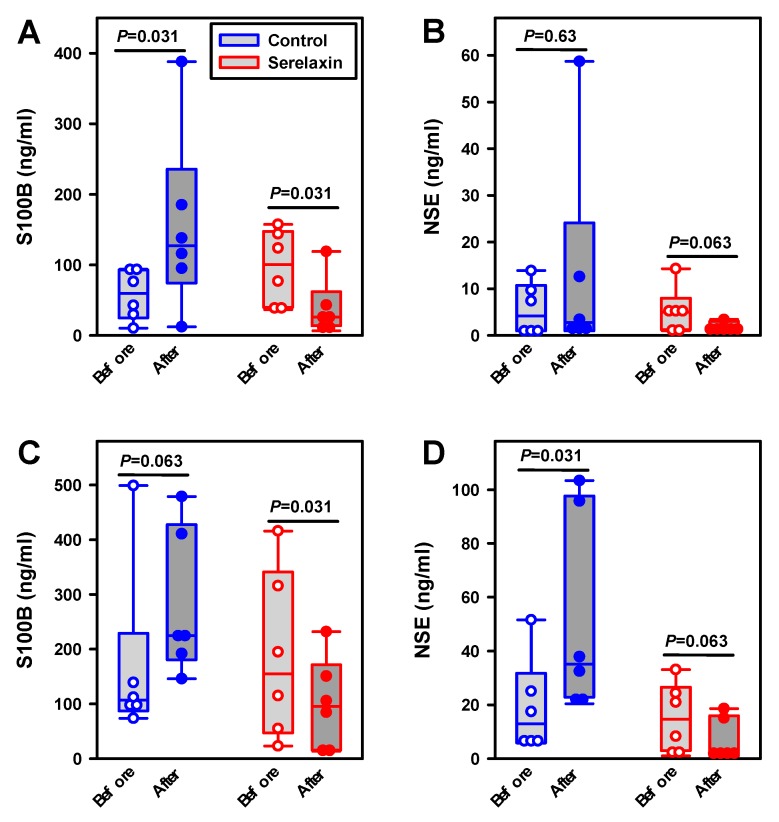
Effects of serelaxin on neuroprotein release into cerebrospinal fluid. Concentrations of neuroproteins S100B and neuron-specific enolase (NSE) were measured before and after experimental interventions. (**A**) S100B during hypoxia, (**B**) NSE during hypoxia, (**C**) S100B during hypovolemia, and (**D**) NSE during hypovolemia. Results are presented as box-plots (boxes represent 25th–75th percentile, with medians and individual values as circles). Comparisons of protein levels before and after interventions were made with Wilcoxon’s signed rank test; *p*-values are indicated in the figure. There were six animals per group.

**Figure 8 ijms-21-01632-f008:**
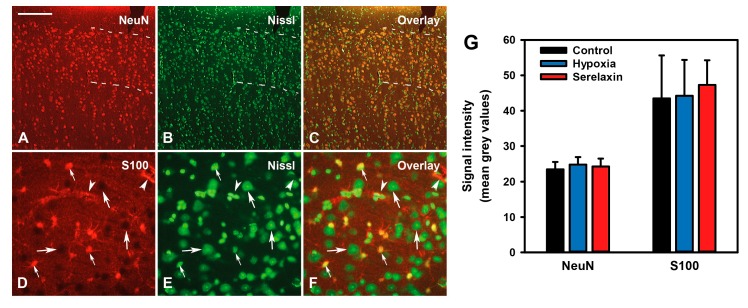
Immunohistochemical labeling of cortical neurons and astrocytes. Transverse cortical sections from a control animal (hypoxia only) stained for the pan-neuronal marker NeuN (**A**) and the astrocyte antigen S100 (**D**). (**B**,**E**) Nissl-like counterstaining and (**C**,**F**) overlays of the respective sections. (**A**–**C**) NeuN expression in neuronal cell bodies; dashed lines delineate cortical layers II and III. (**D**–**F**) S100 labelling of astrocyte cell bodies (thin arrows), astrocyte feet around blood vessels (arrowheads), and astrocyte processes throughout the neurophil. Neuronal cell bodies are S100-negative (thick arrows). Scale bar = 400 µm for A–C, 100 µm for D–F. (**G**) Immunostaining in cortical layers II and III of animals not undergoing the experimental procedure (control) or subjected to hypoxia alone (hypoxia) or hypoxia plus serelaxin treatment (Serelaxin). Mean ± SD; *N* = 5 per group. No differences were found between the groups (ANOVA; NeuN: F_2,12_ = 0.45, *p* = 0.65; S100B: F_2,12_ = 0.21, *p* = 0.82).

**Figure 9 ijms-21-01632-f009:**
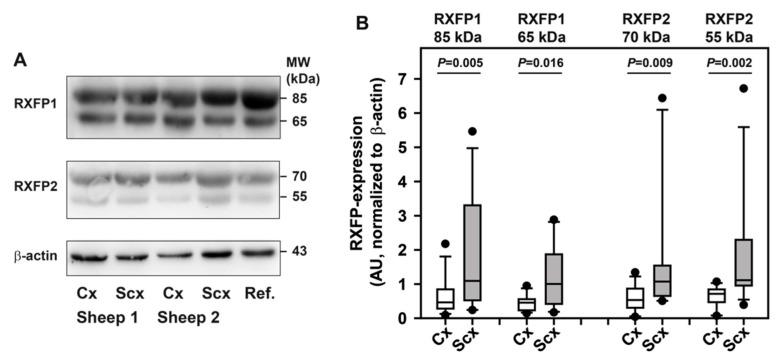
Expression of relaxin receptors in brain vessels. Western blot analysis was performed for cortical and subcortical brain vessels. (**A**) Both RXFP1 and RXFP2 consistently presented as double bands in the cortex and subcortex; data for two sheep are shown. (**B**) Quantitation of receptor expression after normalization of band intensities to the housekeeping protein β-actin showed higher expression levels for all detected bands in the subcortex than in the cortex. As these data were not normally distributed, they are presented as box plots (boxes represent 25th–75th percentile, with medians indicated by a horizontal line). Whiskers indicate the 10th and 90th percentiles, respectively. Dots indicate outliers. Differences were identified with the Wilcoxon signed rank test, *N* = 12 sheep. Cx, cortical vessels; Scx, subcortical vessels; AU, arbitrary units; Ref., reference sample.

**Figure 10 ijms-21-01632-f010:**
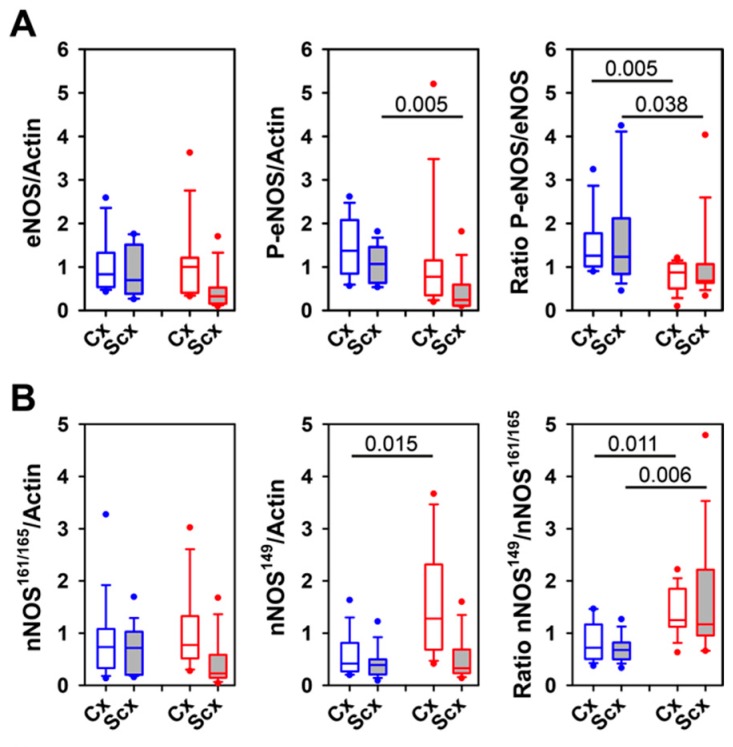
Activation state of downstream relaxin receptor signaling pathways. Western blot analysis was performed for cortical (Cx) and subcortical (Scx) brain vessels of control sheep (blue) and serelaxin-treated sheep (red), both not undergoing the experimental interventions. (**A**) Expression of eNOS and phospho-Ser1177-eNOS normalized to β-actin, and ratio of phospho-Ser1177-eNOS over eNOS. *N* = 12 (control), *N* = 10 (serelaxin). (**B**) Expression of long variants nNOS^161/165^ and truncated nNOS^149^ normalized to β-actin, and ratio of nNOS^149^ to nNOS^161/165^. *N* = 12 (control), *N* = 8 (serelaxin). (**C**) Expression of CREB (cAMP-responsive element-binding protein) variants CREB^60^, CREB^52^, and CREB^43^ normalized to β-actin, and ratio of phospho-Ser133-CREB variants over the respective total protein content. *N* =12 (control), *N* = 10 (serelaxin). (**D**) Expression of ERK1 or ERK2 normalized to β-actin, and ratio of phospho-Thr202/Tyr204-ERK-variants over the respective total protein content. *N* = 8 (control), *N* = 8 (serelaxin). For definitions of box-plots, see legend to Figure 9. Differences were identified with the Wilcoxon signed rank test for the comparison of the cortex and subcortex within a treatment group. The Mann–Whitney rank sum test was used for comparisons between groups.

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
