# Peer review of "Altered Cerebral Blood Flow and Potential Neuroprotective Effect of Human Relaxin-2 (Serelaxin) During Hypoxia or Severe Hypovolemia in a Sheep Model"

_ijms, 2020, doi:10.3390/ijms21051632_

Round 1

Reviewer 1 Report

Comments to the Corresponding Author

Summary:

The manuscript by Schiffner et al. investigated the protective effects of relaxin-2 in hypoxia or hypovolemia in a sheep model. Experiments are well designed and physiological settings are well evaluated. The manuscript is interesting although some concerns were raised regarding the results and statistical analysis.

Major Compulsory Revisions

The biggest problem about this manuscript is that the authors concluded that relaxin-2 had a neuroprotective effect during hypoxia and hypovolemia. Maybe cortical CBF is increasing during hypoxia and hypovolemia by administration of relaxin-2. The authors showed suppression of NSE and S100B levels by administration of relaxin-2 under pathological conditions in the cerebrospinal fluid. However, these are just markers. The authors should prove the hypothesis by in vivo functional scales and/or cell death assay, and functional assay in vitro. At least, their conclusion, improvement of cortical CBF serelaxin might reduce cognitive impairments associated with these emergency settings, is overestimated.

Twelve minutes hypoxia is too short to evaluate the effect of hypoxia. The authors also mentioned the fact. The experiment is not enough to prove the hypothesis.

In Figure 9, the bands of actin were not equal loading. The results did not tell the true answers. In Figure 10, representative WB films were necessary to show the results for densitometric analyses.

Author Response

We thank the referee for her/his critical reading of the manuscript and the helpful suggestions to improve the manuscript.

Comments to the Author

The manuscript by Schiffner et al. investigated the protective effects of relaxin-2 in hypoxia or hypovolemia in a sheep model. Experiments are well designed and physiological settings are well evaluated. The manuscript is interesting although some concerns were raised regarding the results and statistical analysis.

Answer: The linear mixed model calculations are state of the art for our research model. To get unbiased results in regressions models, normal distribution of estimated regression coefficients (and not of the data) is required (Norman, G. (2010). Likert scales, levels of measurement and the “laws” of statistics. Advances in health sciences education, 15(5), 625-632.). Furthermore, it is known that the distribution of estimated regression coefficients is asymptotic normal  (Verbeke, G., & Molenberghs, G. (2009). Linear Mixed Models For Longitudinal Data).

Major Compulsory Revisions

The biggest problem about this manuscript is that the authors concluded that relaxin-2 had a neuroprotective effect during hypoxia and hypovolemia. Maybe cortical CBF is increasing during hypoxia and hypovolemia by administration of relaxin-2. The authors showed suppression of NSE and S100B levels by administration of relaxin-2 under pathological conditions in the cerebrospinal fluid. However, these are just markers. The authors should prove the hypothesis by in vivo functional scales and/or cell death assay, and functional assay in vitro. At least, their conclusion, improvement of cortical CBF serelaxin might reduce cognitive impairments associated with these emergency settings, is overestimated.

Action: Here we partially agree with the reviewer. We measured the changes in the cerebral microcirculation and neuroprotein levels, respectively. But we could not see damage in immunohistochemistry. However, the latter does not mean that there is no neuroprotective effect, as the neuroproteins are extremely sensitive markers for neuronal damage and are associated with cognitive defects.

To better reflect the different levels of (experimental) evidence in our study we changed the title accordingly (See also lines 346-347). Moreover, we added a passage (including several new references) to the discussion, which substantiates our conclusion that relaxin-2 may have a neuroprotective effect based on its improvements of neuroprotein levels.
NSE and S100B are clinically very relevant biomarkers for assessing neuronal damage. See changes in lines 421-425.

Twelve minutes hypoxia is too short to evaluate the effect of hypoxia. The authors also mentioned the fact. The experiment is not enough to prove the hypothesis.    

Answer: Our statement that the 12 minutes of hypoxia are too short for the detection of (massive) morphological signs of pathology should not be interpreted as a flaw in the experimental design. In fact, we clearly see an increased release of neuroproteins as a consequence of hypoxia in the control group.

We slightly changed lines 274-275 to make clear that this sentence only refers to the level of immediate morphological changes.

Figure 9, the bands of actin were not equal loading. The results did not tell the true answers. In Figure 10, representative WB films were necessary to show the results for densitometric analyses.

Answer: We apologize for not being able to present blots with absolute equal loading controls. However, we try to use images for quantitation of the bands, which contain no saturated bands, because saturated band are the real problem for quantitation. We try to use the linear range for signal detection of our imaging system.

In addition, we calculated ratios of the signal intensities (protein x versus actin or phospho-protein x versus protein x) for comparison of different conditions, as described in the methods section. To do this, it is more important to produce images without saturated bands. Furthermore, we included a reference sample in all our blots to be able to compare the results from different membranes. Taken together, using this method in combination with reasonably large sample numbers should provide reliable information about relative protein expression levels in different experimental groups.

Action: We include a complete set of original blot membranes for Figure 10 in the revised submission.

Reviewer 2 Report

Strength:

              In this study, authors tried to clarify the neuroprotective effect of rekaxin-2 during hypoxia and hypovolemia animal model in sheep. The treatment of relaxin-2 could increase CBF and subcortical CBF in hypoxia animal model and CBF in hypovolemia animal model. In addition, the concentrations both S100B and NSE in cerebrospinal fluid were decreased by the treatment of relaxin-2 compared to before treatment or control group. In addition, western blot analysis revealed that there was region specific difference in the expression of relaxin receptor levels and the relaxin- related signaling. Considering that, relaxin-2 might have neuroprotective effect.

Comment:

The treatment of relaxin-2 could increase CBF and subcortical CBF in hypoxia animal model. In contrast, the treatment of relaxin-2 could increase CBF and decrease subcortical CBF in hypovolemia animal model. Authors need to discuss why this phenomenon emerged with mentioning the different effect of relaxin-2 in its signaling cascade between CBF and subcortical CBF more in detail.

In Fig.9 and Fig.10, all samples were obtained from animals which were not treated with the experimental interventions. It seems that relaxin-2 did not show any significant effect except for the conditions under both hypoxia and hypovolemia according to Fig.1-6. In other words, relaxin-2 may not affect to basal condition. If so, does the analysis of signal cascades of relaxin-2 in animals which were not treated with the experimental interventions reflect biological properties under hypoxia and hypovolemia?

In Fig.7, please correct the graph labels in horizontal axis.

In Fig.9, “beta” of beta-actin is miss-converted.

Author Response

We thank the referee for her/his critical reading of the manuscript and the helpful suggestions to improve the manuscript.

Comments to the Author

Strength:

In this study, authors tried to clarify the neuroprotective effect of rekaxin-2 during hypoxia and hypovolemia animal model in sheep. The treatment of relaxin-2 could increase CBF and subcortical CBF in hypoxia animal model and CBF in hypovolemia animal model. In addition, the concentrations both S100B and NSE in cerebrospinal fluid were decreased by the treatment of relaxin-2 compared to before treatment or control group. In addition, western blot analysis revealed that there was region specific difference in the expression of relaxin receptor levels and the relaxin- related signaling. Considering that, relaxin-2 might have neuroprotective effect.

Comment:

The treatment of relaxin-2 could increase CBF and subcortical CBF in hypoxia animal model. In contrast, the treatment of relaxin-2 could increase CBF and decrease subcortical CBF in hypovolemia animal model. Authors need to discuss why this phenomenon emerged with mentioning the different effect of relaxin-2 in its signaling cascade between CBF and subcortical CBF more in detail.

Answer: We presently can only speculate about the precise mechanism(s) involved. The unexpectedly higher RXFP-expression in subcortical vessels might explain the differential effects (but we tried to avoid too much speculation in the manuscript). Among other explanations, one interesting explanation could be that higher RXFP-expression might be the basis for the better blood-supply of the subcortex under “stress” conditions. That would include presence of a basal amount of RXFP-ligands, which could be sufficient to trigger an appropriate response in subcortical vessels. Exogenous relaxin-2 could lead to a maximum stimulation of the cortical vessels and thus provide better blood supply. However, as we presently have no experimental data clearly supporting this, we refrained from including this in the discussion.

As from our analysis of signaling pathways no highly likely mechanism for differential responses in cortical versus subcortical vessels can be deduced, we would like to refrain from extending the discussion, because it would become too speculative. However, this issue should be analyzed in more detail (please see next answer).

In Fig.9 and Fig.10, all samples were obtained from animals which were not treated with the experimental interventions. It seems that relaxin-2 did not show any significant effect except for the conditions under both hypoxia and hypovolemia according to Fig.1-6. In other words, relaxin-2 may not affect to basal condition. If so, does the analysis of signal cascades of relaxin-2 in animals which were not treated with the experimental interventions reflect biological properties under hypoxia and hypovolemia?

Answer/Action: The limitations of using animals not undergoing the experimental procedures for the analysis of RXFPs and downstream signaling proteins are obvious. We think that this fact is clearly communicated in the discussion. A potentially misleading phrase has been corrected (lines 462-463).

Analyzing signaling pathways in these animals nevertheless revealed that there exist differences between cortex and subcortex regarding relaxin-2-induced signaling. To show this was our primary intention when we planned these experiments.

It is totally clear that more detailed analyses are necessary to elucidate the role of relaxin-2, its receptors and the signaling mechanisms. However, this was not planned when starting the authorization procedure for our experiments under the german animal rights law.

For a detailed analysis a considerably higher number of animals will have to undergo the experimental procedures, because samples have to be taken at several time-points. Authorization for this number of animals has to be justified. Therefore, we included the available data in this manuscript – they should provide the necessary evidence that doing a detailed analysis is warranted.

In Fig.7, please correct the graph labels in horizontal axis.

Answer/Action: The labels are visible in the submitted files and in the Word-file we received for revision. We are not sure at what stage of file conversion these labels vanished.

We will provide the original files separately for inclusion in the final version of the manuscript and we will turn our attention to this issue during proofreading.

In Fig.9, “beta” of beta-actin is miss-converted.

Answer: Correct symbol is included.

Reviewer 3 Report

Schiffner et al. describe that Serelaxin, a nitric oxide-dependent vasodilator, is neuroprotective after hypoxia and hypovolemia via increasing cortical cerebral blood flow. In addition, they showed that concentrations of neuroproteins S100B and NSE were reduced in serelaxin-treated group compared to control group after both interventions.

Major comments:

The authors claim that both hypovolemia and hypoxia induce neuronal damage based on increased concentrations of neuroproteins S100B and NSE. However, Figure 7 show that values after hypoxia in the control group (Figure 7A) are not different to values before hypovolemia in the same group (Figure 7C). In addition, immunohistoligical analysis of brain sections failed to demonstrate any neuronal damage after hypoxia. Therefore, there is not sufficient evidence to claim that hypoxia induced neuronal damage. In Figure 8. Immunohistological analysis of brains after hypoxia are shown, however it will be interesting to perform the same analysis after hypovolemia. The reduction in cerebral blood flow is more severe during 50% blood removal, which may induce more severe brain damage and may be detected by immunohistological markers. Figure 10.The authors claim that signal transduction pathways rather than receptor expression are the responsible of the increase cortical CBF in serelaxin treatment group, however the analysis of the signaling pathways should be performed after hypoxia and hypovolemia to address this point. How the authors explain that serelaxin receptor expression was higher in subcortical vessel vs cortical? This is unexpected considering that serelaxin treatment increases cortical CBF?

Minor comments:

Line 56 “Initial hopes that serelaxin might be beneficial in acute heart failure scatterd” please correct

Author Response

We thank the referee for her/his critical reading of the manuscript and the helpful suggestions to improve the manuscript.

Comments to the Author

Schiffner et al. describe that Serelaxin, a nitric oxide-dependent vasodilator, is neuroprotective after hypoxia and hypovolemia via increasing cortical cerebral blood flow. In addition, they showed that concentrations of neuroproteins S100B and NSE were reduced in serelaxin-treated group compared to control group after both interventions.

The authors claim that both hypovolemia and hypoxia induce neuronal damage based on increased concentrations of neuroproteins S100B and NSE. However, Figure 7 show that values after hypoxia in the control group (Figure 7A) are not different to values before hypovolemia in the same group (Figure 7C). In addition, immunohistoligical analysis of brain sections failed to demonstrate any neuronal damage after hypoxia. Therefore, there is not sufficient evidence to claim that hypoxia induced neuronal damage. In Figure 8. Immunohistological analysis of brains after hypoxia are shown, however it will be interesting to perform the same analysis after hypovolemia. The reduction in cerebral blood flow is more severe during 50% blood removal, which may induce more severe brain damage and may be detected by immunohistological markers.

Answer/Action: We partially agree with the reviewer. We measured the changes in the cerebral microcirculation and neuroprotein levels, respectively. But we could not see damage in immunohistochemistry. However, the latter does not mean that there is no neuroprotective effect, as the neuroproteins are extremely sensitive markers for neuronal damage and are associated with cognitive defects.

The differences in baseline release of neuroproteins as summarized in Fig. 7 is the result of several factors. First, there are inter-individual responses of the animals (therefore, we used the Wilcoxon signed rank test for comparisons). Second, the first sampling for neuroprotein measurements was done immediately before the experimental intervention, i.e. blood removal or oxygen withdrawal. That means the “before” sampling took place after preparation of the animals, craniectomy, insertion of the probes, etc. – which are responsible for baseline neuroprotein release. Sampling time was chosen to allow detection of changes induced by hypovolemia or hypoxia, respectively.

To better reflect the different levels of (experimental) evidence in our study we changed the title accordingly (See also lines 346-347). Moreover, we added a passage (including several new references) to the discussion, which substantiates our conclusion that relaxin-2 may have a neuroprotective effect based on its improvements of neuroprotein levels.

NSE and S100B are clinically very relevant biomarkers for assessing neuronal damage. See changes in lines 421-425. In addition, we slightly changed lines 274-275 to make clear that this sentence only refers to the level of immediate morphological changes.

Concerning the severity of detectable neuronal damage in both models, we do not expect to find significant differences between experimental groups. The expected numbers of damaged cells are to low to be significantly higher than the numbers of cells damaged during sample preparation and fixation for immunohistochemistry. We would not expect to see 5 % dead cells after a short time hypovolemia or hypoxia.

Figure 10.The authors claim that signal transduction pathways rather than receptor expression are the responsible of the increase cortical CBF in serelaxin treatment group, however the analysis of the signaling pathways should be performed after hypoxia and hypovolemia to address this point. How the authors explain that serelaxin receptor expression was higher in subcortical vessel vs cortical? This is unexpected considering that serelaxin treatment increases cortical CBF?

Answer/Action: The limitations of using animals not undergoing the experimental procedures for the analysis of RXFPs and downstream signaling proteins are obvious. We think that this fact is clearly communicated in the discussion. A potentially misleading phrase has been corrected (lines 462-463).

Analyzing signaling pathways in these animals nevertheless revealed that there exist differences between cortex and subcortex regarding relaxin-2-induced signaling. To show this was our primary intention when we planned these experiments.

It is totally clear that more detailed analyses are necessary to elucidate the role of relaxin-2, its receptors and the signaling mechanisms. However, this was not planned when starting the authorization procedure for our experiments under the german animal rights law.

For a detailed analysis a considerably higher number of animals will have to undergo the experimental procedures, because samples have to be taken at several time-points. Authorization for this number of animals has to be justified. Therefore, we included the available data in this manuscript – they should provide the necessary evidence that doing a detailed analysis is warranted.

We presently can only speculate about the unexpectedly higher RXFP-expression in subcortical vessels (and tried to avoid this in the manuscript). Among other explanations, one interesting explanation could be that higher RXFP-expression might be the basis for the better blood-supply of the subcortex under “stress” conditions. That would include presence of a basal amount of RXFP-ligands, which could be sufficient to trigger an appropriate response in subcortical vessels. Exogenous relaxin-2 could lead to a maximum stimulation of the cortical vessels and thus provide better blood supply. However, as we presently have no experimental data clearly supporting this, we refrained from including this in the discussion.

Line 56 “Initial hopes that serelaxin might be beneficial in acute heart failure scatterd” please correct

Answer/Action: We altered this passage of the introduction.

Round 2

Reviewer 1 Report

The reviewer does not have any comments about the present version.

Author Response

We thank the reviewer for his revision.

Reviewer 3 Report

This revised manuscript does not address the major concern of the reviewer, which is the lack of robust evidence regarding the neuroprotective effect of Serelaxin. Also, the authors' answer regarding this point ("That means the “before” sampling took place after preparation of the animals, craniectomy, insertion of the probes, etc. – which are responsible for baseline neuroprotein release") is misleading since the preparation procedure in the brain (i.e. craniectomy) should be the same for hypoxia and hypovolemia. A sham-operated group should be included in the study design.

Author Response

We thank the referee for her/his critical reading of the manuscript and the helpful suggestions to improve the manuscript.

Comments to the Author

This revised manuscript does not address the major concern of the reviewer, which is the lack of robust evidence regarding the neuroprotective effect of Serelaxin. Also, the authors' answer regarding this point ("That means the “before” sampling took place after preparation of the animals, craniectomy, insertion of the probes, etc. – which are responsible for baseline neuroprotein release") is misleading since the preparation procedure in the brain (i.e. craniectomy) should be the same for hypoxia and hypovolemia. A sham-operated group should be included in the study design.

Answer/Action: We changed the Title, lines 41-42 of the abstract, and lines 348 and 426, respectively, of the discussion.

As detailed in round 1 we partially agree with reviewer 3. We, therefore had modified the title to better express the potential neuroprotective effec to reflect the lack of robust direct evidence for a neuroprotective effect of serelaxin (in our animal model). However, we uphold our statement in the first response regarding the validity of neuroproteins as markers for neuronal damage, which we had included in the discussion.

We furthermore rephrased the sentence in lines 41-42 (abstract’s penultimate sentence) from “Our findings support the hypothesis that serelaxin has a neuroprotective effect during hypoxia and hypovolemia” to "Our findings support the hypothesis that serelaxin is a potential neuroprotectant during hypoxia and hypovolemia”.

Similarly in lines 348 and 426 (Discussion), we changed the “strongly suggest” to “imply”.

Round 3

Reviewer 3 Report

The manuscript has been substantially improved and there are no further comments.